# FOCAL: Efficient Fully-Offline Meta-Reinforcement Learning via Distance Metric Learning and Behavior Regularization

**Lanqing Li**[1,*] **Rui Yang**[2,†] **Dijun Luo**[1]
[1] Tencent AI Lab
[2] Department of Automation, Tsinghua University
`lanqingli1993@gmail.com, dijunluo@tencent.com`
`yangrui19@mails.tsinghua.edu.cn`

## ABSTRACT

We study the offline meta-reinforcement learning (OMRL) problem, a paradigm which enables reinforcement learning (RL) algorithms to quickly adapt to unseen tasks without any interactions with the environments, making RL truly practical in many real-world applications. This problem is still not fully understood, for which two major challenges need to be addressed. First, offline RL usually suffers from bootstrapping errors of out-of-distribution state-actions which leads to divergence of value functions. Second, meta-RL requires efficient and robust task inference learned jointly with control policy. In this work, we enforce behavior regularization on learned policy as a general approach to offline RL, combined with a deterministic context encoder for efficient task inference. We propose a novel negative-power distance metric on bounded context embedding space, whose gradients propagation is detached from the Bellman backup. We provide analysis and insight showing that some simple design choices can yield substantial improvements over recent approaches involving meta-RL and distance metric learning. To the best of our knowledge, our method is the first model-free and end-to-end OMRL algorithm, which is computationally efficient and demonstrated to outperform prior algorithms on several meta-RL benchmarks.[1]

## 1 INTRODUCTION

Applications of reinforcement learning (RL) in real-world problems have been proven successful in many domains such as games (Silver et al., 2017; Vinyals et al., 2019; Ye et al., 2020) and robot control (Johannink et al., 2019). However, the implementations so far usually rely on interactions with either real or simulated environments. In other areas like healthcare (Gottesman et al., 2019), autonomous driving (Shalev-Shwartz et al., 2016) and controlled-environment agriculture (Binas et al., 2019) where RL shows promise conceptually or in theory, exploration in real environments is evidently risky, and building a high-fidelity simulator can be costly. Therefore a key step towards more practical RL algorithms is the ability to learn from static data. Such paradigm, termed "offline RL" or "batch RL", would enable better generalization by incorporating diverse prior experience. Moreover, by leveraging and reusing previously collected data, off-policy algorithms such as SAC (Haarnoja et al., 2018) has been shown to achieve far better sample efficiency than on-policy methods. The same applies to offline RL algorithms since they are by nature off-policy.

The aforementioned design principles motivated a surge of recent works on offline/batch RL (Fujimoto et al., 2019; Kumar et al., 2019; Wu et al., 2019; Siegel et al., 2020). These papers propose remedies by regularizing the learner to stay close to the logged transitions of the training datasets, namely the behavior policy, in order to mitigate the effect of bootstrapping error (Kumar et al., 2019), where evaluation errors of out-of-distribution state-action pairs are never corrected and hence easily diverge due to inability to collect new data samples for feedback. There exist claims that offline RL

---

[*]Correspondence to: Lanqing Li <lanqingli1993@gmail.com>, Dijun Luo <dijunluo@tencent.com>
[†]Work done while an intern at Tencent AI Lab.

[1]Source code: `https://github.com/FOCAL-ICLR/FOCAL-ICLR/`

can be implemented successfully without explicit correction for distribution mismatch given sufficiently large and diverse training data (Agarwal et al., 2020). However, we find such assumption unrealistic in many practices, including our experiments. In this paper, to tackle the out-of-distribution problem in offline RL in general, we adopt the proposal of behavior regularization by Wu et al. (2019).

For practical RL, besides the ability to learn without exploration, it's also ideal to have an algorithm that can generalize to various scenarios. To solve real-world challenges in multi-task setting, such as treating different diseases, driving under various road conditions or growing diverse crops in autonomous greenhouses, a robust agent is expected to quickly transfer and adapt to unseen tasks, especially when the tasks share common structures. Meta-learning methods (Vilalta & Drissi, 2002; Thrun & Pratt, 2012) address this problem by learning an inductive bias from experience collected across a distribution of tasks, which can be naturally extended to the context of reinforcement learning. Under the umbrella of this so-called meta-RL, almost all current methods require on-policy data during either both meta-training and testing phases (Wang et al., 2016; Duan et al., 2016; Finn et al., 2017) or at least testing stage (Rakelly et al., 2019) for adaptation. An efficient and robust method which incorporates both fully-offline learning and meta-learning in RL, despite few attempts (Li et al., 2019b; Dorfman & Tamar, 2020), has not been fully developed and validated.

In this paper, under the first principle of maximizing practicality of RL algorithm, we propose an efficient method that integrates task inference with RL algorithms in a fully-offline fashion. Our fully-offline context-based actor-critic meta-RL algorithm, or FOCAL, achieves excellent sample efficiency and fast adaptation with limited logged experience, on a range of deterministic continuous control meta-environments. The primary contribution of this work is designing the first end-to-end and model-free offline meta-RL algorithm which is computationally efficient and effective without any prior knowledge of task identity or reward/dynamics. To achieve efficient task inference, we propose an inverse-power loss for effective learning and clustering of task latent variables, in analogy to coulomb potential in electromagnetism, which is also unseen in previous work. We also shed light on the specific design choices customized for OMRL problem by theoretical and empirical analyses.

## 2 RELATED WORK

**Meta-RL**    Our work FOCAL builds upon the meta-learning framework in the context of reinforcement learning. Among all paradigms of meta-RL, this paper is most related to the context-based and metric-based approaches. Context-based meta-RL employs models with memory such as recurrent (Duan et al., 2016; Wang et al., 2016; Fakoor et al., 2019), recursive (Mishra et al., 2017) or probabilistic (Rakelly et al., 2019) structures to achieve fast adaptation by aggregating experience into a latent representation on which the policy is conditioned. The design of the context usually leverages the temporal or Markov properties of RL problems.

Metric-based meta-RL focuses on learning effective task representations to facilitate task inference and conditioned control policies, by employing techniques such as distance metric learning (Yang & Jin, 2006). Koch et al. (2015) proposed the first metric-based meta-algorithm for few-shot learning, in which a Siamese network (Chopra et al., 2005) is trained with triplet loss to compare the similarity between a query and supports in the embedding space. Many metric-based meta-RL algorithms extend these works (Snell et al., 2017; Sung et al., 2018; Li et al., 2019a).

Among all aforementioned meta-learning approaches, this paper is most related to the context-based PEARL algorithm (Rakelly et al., 2019) and metric-based prototypical networks (Snell et al., 2017). PEARL achieves SOTA performance for off-policy meta-RL by introducing a probabilistic permutation-invariant context encoder, along with a design which disentangles task inference and control by different sampling strategies. However, it requires exploration during meta-testing. The prototypical networks employ similar design of context encoder as well as an Euclidean distance metric on deterministic embedding space, but tackles meta-learning of classification tasks with squared distance loss as opposed to the inverse-power loss in FOCAL for the more complex OMRL problem.

**Offline/Batch RL**    To address the bootstrapping error (Kumar et al., 2019) problem of offline RL, this paper adopts behavior regularization directly from Wu et al. (2019), which provides a relatively unified framework of several recent offline or off-policy RL methods (Haarnoja et al.,

2018; Fujimoto et al., 2019; Kumar et al., 2019). It incorporates a divergence function between distributions over state-actions in the actor-critic objectives. As with SAC (Haarnoja et al., 2018), one limitation of the algorithm is its sensitivity to reward scale and regularization strength. In our experiments, we indeed observed wide spread of optimal hyper-parameters across different meta-RL environments, shown in Table 4.

**Offline Meta-RL**    To the best of our knowledge, despite attracting more and more attention, the offline meta-RL problem is still understudied. We are aware of a few papers that tackle the same problem from different angles (Li et al., 2019b; Dorfman & Tamar, 2020). Li et al. (2019b) focuses on a specific scenario where biased datasets make the task inference module prone to overfit the state-action distributions, ignoring the reward/dynamics information. This so-called MDP ambiguity problem occurs when datasets of different tasks do not have significant overlap in their state-action visitation frequencies, and is exacerbated by sparse rewards. Their method MBML requires training of offline BCQ (Fujimoto et al., 2019) and reward/dynamics models for each task, which are computationally demanding, whereas our method is end-to-end and model-free.

Dorfman & Tamar (2020) on the other hand, formulate the OMRL as a Bayesian RL (Ghavamzadeh et al., 2016) problem and employs a probabilistic approach for Bayes-optimal exploration. Therefore we consider their methodology tangential to ours.

## 3    PRELIMINARIES

### 3.1    NOTATIONS AND PROBLEM STATEMENT

We consider fully-observed Markov Decision Process (MDP) (Puterman, 2014) in deterministic environments such as MuJoCo (Todorov et al., 2012). An MDP can be modeled as $\mathcal{M} = (\mathcal{S}, \mathcal{A}, P, R, \rho_0, \gamma)$ with state space $\mathcal{S}$, action space $\mathcal{A}$, transition function $P(s'|s, a)$, bounded reward function $R(s, a)$, initial state distribution $\rho_0(s)$ and discount factor $\gamma \in (0, 1)$. The goal is to find a policy $\pi(a|s)$ to maximize the cumulative discounted reward starting from any state. We introduce the notion of multi-step state marginal of policy $\pi$ as $\mu_\pi^t(s)$, which denotes the distribution over state space after rolling out $\pi$ for $t$ steps starting from state $s$. The notation $R_\pi(s)$ denotes the expected reward at state $s$ when following policy $\pi$: $R_\pi(s) = \mathbb{E}_{a \sim \pi}[R(s, a)]$. The state-value function (a.k.a. value function) and action-value function (a.k.a Q-function) are therefore

$$V_\pi(s) = \sum_{t=0}^{\infty} \gamma^t \mathbb{E}_{s_t \sim \mu_\pi^t(s)}[R(s_t)] \tag{1}$$

$$Q_\pi(s, a) = R(s, a) + \gamma \mathbb{E}_{s' \sim P(s'|s, a)}[V_\pi(s')] \tag{2}$$

Q-learning algorithms are implemented by iterating the Bellman optimality operator $\mathcal{B}$, defined as:

$$(\mathcal{B}\hat{Q})(s, a) := R(s, a) + \gamma \mathbb{E}_{P(s'|s, a)}[\max_{a'} \hat{Q}(s', a')] \tag{3}$$

When the state space is large/continuous, $\hat{Q}$ is used as a hypothesis from the set of function approximators (e.g. neural networks).

In the offline context of this work, given a distribution of tasks $p(\mathcal{T})$ where every task is an MDP, we study off-policy meta-learning from collections of static datasets of transitions $\mathcal{D}_i = \{(s_{i,t}, a_{i,t}, s'_{i,t}, r_{i,t})|t = 1, ..., N\}$ generated by a set of behavior policies $\{\beta_i(a|s)\}$ associated with each task index $i$. A key underlying assumption of meta-learning is that the tasks share some common structures. By definition of MDP, in this paper we restrict our attention to tasks with shared state and action space, but differ in transition and reward functions.

We define the meta-optimization objective as

$$\mathcal{L}(\theta) = \mathbb{E}_{\mathcal{T}_i \sim p(\mathcal{T})}\left[\mathcal{L}_{\mathcal{T}_i}(\theta)\right] \tag{4}$$

where $\mathcal{L}_{\mathcal{T}_i}(\theta)$ is the objective evaluated on transition samples drawn from task $\mathcal{T}_i$. A common choice of $p(\mathcal{T})$ is the uniform distribution on the set of given tasks $\{\mathcal{T}_i|i = 1, ..., n\}$. In this case, the meta-training procedure turns into minimizing the average losses across all training tasks

$$\hat{\theta}_{\text{meta}} = \arg\min_\theta \frac{1}{n} \sum_{k=1}^{n} \mathbb{E}\left[\mathcal{L}_k(\theta)\right] \tag{5}$$

### 3.2 Behavior Regularized Actor Critic (BRAC)

Similar to SAC, to constrain the bootstrapping error in offline RL, for each individual task $\mathcal{T}_i$, behavior regularization (Wu et al., 2019) introduces a divergence measure between the learner $\pi_\theta$ and the behavior policy $\pi_b$ in value and target Q-functions. For simplicity, we ignore task index in this section:

$$V_\pi^D(s) = \sum_{t=0}^\infty \gamma^t \mathbb{E}_{s_t \sim \mu_\pi^t(s)} \left[ R_\pi(s_t) - \alpha D(\pi_\theta(\cdot|s_t), \pi_b(\cdot|s_t)) \right] \tag{6}$$

$$\bar{Q}_\psi^D(s,a) = \bar{Q}_\psi(s,a) - \gamma\alpha\hat{D}(\pi_\theta(\cdot|s), \pi_b(\cdot|s)) \tag{7}$$

where $\bar{Q}$ denotes a target Q-function without gradients and $\hat{D}$ denotes a sample-based estimate of the divergence function $D$. In actor-critic framework, the loss functions of Q-value and policy learning are given by, respectively,

$$\mathcal{L}_{critic} = \mathbb{E}_{\substack{(s,a,r,s')\sim\mathcal{D} \\ a'\sim\pi_\theta(\cdot|s')}} \left[ \left( r + \gamma\bar{Q}_\psi^D(s',a') - Q_\psi(s,a) \right)^2 \right] \tag{8}$$

$$\mathcal{L}_{actor} = -\mathbb{E}_{(s,a,r,s')\sim\mathcal{D}} \left[ \mathbb{E}_{a''\sim\pi_\theta(\cdot|s)}[Q_\psi(s,a'')] - \alpha\hat{D} \right] \tag{9}$$

### 3.3 Context-Based meta-RL

Context-based meta-RL algorithms aggregate context information, typically in form of task-specific transitions, into a latent space $\mathcal{Z}$. It can be viewed as a special form of RL on partially-observed MDP (Kaelbling et al., 1998) in which a latent representation $z$ as the unobserved part of the state needs to be inferred. Once given complete information of $z$ and $s$ combined as the full state, the learning of the universal policy $\pi_\theta(s,z)$ and value function $V_\pi(s,z)$ (Schaul et al., 2015) becomes RL on regular MDP, and properties of regular RL such as the existence of optimal policy and value functions hold naturally. We therefore formulate the context-based meta-RL problem as solving a *task-augmented MDP* (TA-MDP). The formal definitions are provided in Appendix B.

## 4 Method

Based on our formulation of context-based meta-RL problem, FOCAL first learns an effective representation of meta-training tasks on latent space $\mathcal{Z}$, then solves the offline RL problem on TA-MDP with behavior regularized actor critic method. We illustrate our training procedure in Figure 1 and describe the detailed algorithm in Appendix A. We assume that pre-collected datasets are available for both training and testing phases, making our algorithm fully offline. Our method consists of three key design choices: deterministic context encoder, distance metric learning on latent space as well as decoupled training of task inference and control.

### 4.1 Deterministic Context Encoder

Similar to Rakelly et al. (2019), we introduce an inference network $q_\phi(z|c)$, parameterized by $\phi$, to infer task identity from context $c \sim \mathcal{C}$. In terms of the context encoder design, recent meta-RL methods either employ recurrent neural networks (Duan et al., 2016; Wang et al., 2016) to capture the temporal correlation, or use probabilistic models (Rakelly et al., 2019) for uncertainties estimation. These design choices are proven effective in on-policy and partially-offline off-policy algorithms. However, since our approach aims to address the fully-offline meta-RL problem, we argue that a deterministic context encoder works better in this scenario, given a few assumptions:

First, we consider only *deterministic MDP* in this paper, where the transition function is a Dirac delta distribution. We assume that all meta-learning tasks in this paper are deterministic MDPs, which is satisfied by common RL benchmarks such as MuJoCo. The formal definitions are detailed in Appendix B. Second, we assume all tasks share the same state and action space, while each is characterized by a unique combination of transition and reward functions. Mathematically, this means there exists an injective function $f : \mathcal{T} \to \mathcal{P} \times \mathcal{R}$, where $\mathcal{P}$ and $\mathcal{R}$ are functional spaces of transition probability $P : \mathcal{S} \times \mathcal{A} \times \mathcal{S} \to \{0,1\}$ and bounded reward $R : \mathcal{S} \times \mathcal{A} \to \mathbb{R}$ respectively. A

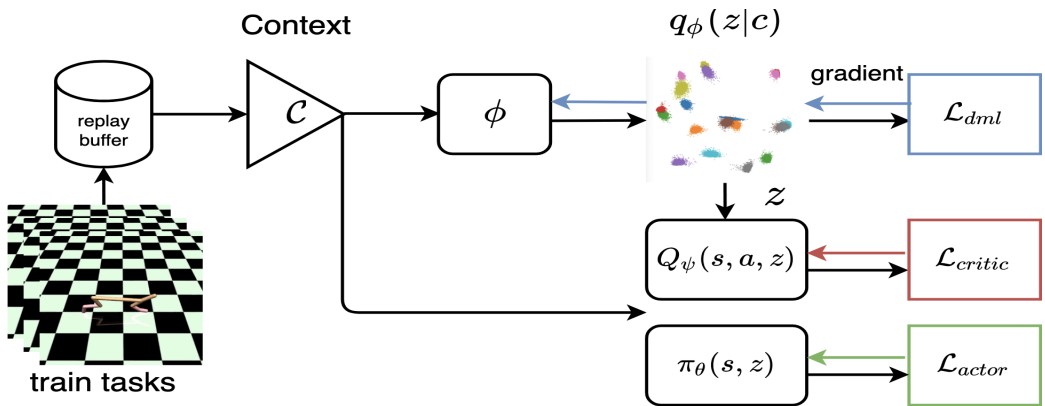

Figure 1: **Meta-training procedure.** The inference network $q_\phi$ uses context data $c$ to compute the latent context variable $z$, which conditions the actor and critic, and is optimized by the distance metric learning (DML) objective. The learning of context encoder ($\mathcal{L}_{dml}$) and control policy ($\mathcal{L}_{actor}$, $\mathcal{L}_{critic}$) are decoupled in terms of gradients.

stronger condition of this injective property is that for any state-action pair $(s, a)$, the corresponding transition and reward are *point-wise unique* across all tasks, which brings the following assumption:

**Assumption 1 (Task-Transition Correspondence).** *We consider meta-RL with a task distribution $p(\mathcal{T})$ to satisfy task-transition correspondence if and only if $\forall \mathcal{T}_1, \mathcal{T}_2 \sim p(\mathcal{T})$, $(s, a) \in \mathcal{S} \times \mathcal{A}$:*

$$P_1(\cdot|s, a) = P_2(\cdot|s, a), R_1(s, a) = R_2(s, a) \iff \mathcal{T}_1 = \mathcal{T}_2 \qquad (10)$$

Under the deterministic MDP assumption, the transition probability function $P(\cdot|s, a)$ is associated with the transition map $t : \mathcal{S} \times \mathcal{A} \to \mathcal{S}$ (Definition B.3). The task-transition correspondence suggests that, given the action-state pair $(s, a)$ and task $\mathcal{T}$, there exists a unique transition-reward pair $(s', r)$. Based on these assumptions, one can define a task-specific map $f_\mathcal{T} : \mathcal{S} \times \mathcal{A} \to \mathcal{S} \times \mathbb{R}$ on the set of transitions $\mathcal{D}_\mathcal{T}$:

$$f_\mathcal{T}(s_t, a_t) = (s'_t, r_t), \quad \forall \mathcal{T} \sim p(\mathcal{T}), (s_t, a_t, s'_t, r_t) \in \mathcal{D}_\mathcal{T} \qquad (11)$$

Recall that all tasks defined in this paper share the same state-action space, hence $\{f_\mathcal{T} | \mathcal{T} \sim p(\mathcal{T})\}$ forms a function family defined on the transition space $\mathcal{S} \times \mathcal{A} \times \mathcal{S} \times \mathbb{R}$, which is also by definition the context space $\mathcal{C}$. This lends a new interpretation that as a task inference module, the context encoder $q_\phi(z|c)$ enforces an embedding of the task-specific map $f_\mathcal{T}$ on the latent space $\mathcal{Z}$, i.e. $q_\phi : \mathcal{S} \times \mathcal{A} \times \mathcal{S} \times \mathbb{R} \to \mathcal{Z}$. Following Assumption 1, every transition $\{s_i, a_i, s'_i, r_i\}$ corresponds to a unique task $\mathcal{T}_i$, which means in principle, task identity can be inferred from any single transition tuple. This implies the context encoder should be permutation-invariant and deterministic, since the embedding of context does not depend on the order of the transitions nor involve any uncertainty. This observation is crucial since it provides theoretical basis for few-shot learning (Snell et al., 2017; Sung et al., 2018) in our settings. In particular, when learning in a fully-offline fashion, any meta-RL algorithm at test-time cannot perform adaptation by exploration. The theoretical guarantee that a few randomly-chosen transitions can enable effective task inference ensures that FOCAL is feasible and efficient.

## 4.2 Distance Metric Learning (DML) of Latent Variables

In light of our analysis on the context encoder design, the goal of task inference is to learn a robust and effective representation of context for better discrimination of task identities. Unlike PEARL, which requires Bellman gradients to train the inference network, our insight is to disentangle the learning of context encoder from the learning of control policy. As explained in previous reasoning about the deterministic encoder, the latent variable is a representation of the task properties involving only dynamics and reward, which in principle should be completely captured by the transition datasets. Given continuous neural networks as function approximators, the learned value functions conditioned on latent variable $z$ cannot distinguish between tasks if the corresponding embedding vectors are too close (Appendix C). Therefore for implementation, we formulate the latent variable learning problem as obtaining the embedding $q_\phi : \mathcal{S} \times \mathcal{A} \times \mathcal{S} \times \mathbb{R} \to \mathcal{Z}$ of transition data

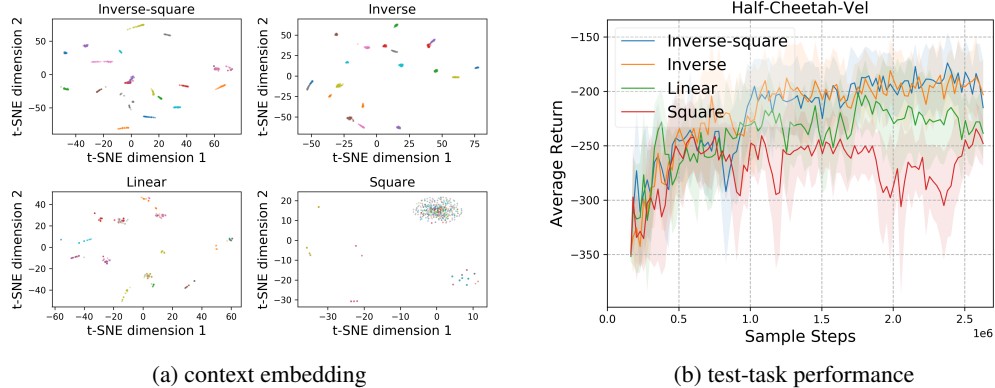

(a) context embedding        (b) test-task performance

Figure 2: (a) t-SNE visualization of embedding vectors drawn from 20 randomized tasks on Half-Cheetah-Vel. Inverse-power distance metric losses (DML) achieve better clustering. Data points are color-coded according to task identity. (b) FOCAL trained with inverse-power DML losses outperform the linear and square distance losses.

$\mathcal{D}_i = \{(s_{i,t}, a_{i,t}, s'_{i,t}, r_{i,t}) | t = 1, ..., N\}$ that clusters similar data (same task) while pushes away dissimilar samples (different tasks) on the embedding space $\mathcal{Z}$, which is essentially distance metric learning (DML) (Sohn, 2016). A common loss function in DML is contrasitive loss (Chopra et al., 2005; Hadsell et al., 2006). Given input data $\boldsymbol{x_i}, \boldsymbol{x_j} \in \mathcal{X}$ and label $y \in \{1, ..., L\}$, it is written as

$$\mathcal{L}^m_{cont}(\boldsymbol{x_i}, \boldsymbol{x_j}; q) = \mathbf{1}\{y_i = y_j\}||\boldsymbol{q_i} - \boldsymbol{q_j}||^2_2 + \mathbf{1}\{y_i \neq y_j\}\max(0, m - ||\boldsymbol{q_i} - \boldsymbol{q_j}||_2)^2 \quad (12)$$

where $m$ is a constant parameter, $\boldsymbol{q_i} = q_\phi(\boldsymbol{x_i})$ is the embedding vector of $\boldsymbol{x_i}$. For data point of different tasks/labels, contrastive loss rewards the distance between their embedding vectors by $L^2$ norm, which is weak when the distance is small, as in the case when $z$ is normalized and $q_\phi$ is randomly initialized. Empirically, we observe that objectives with positive powers of distance lead to degenerate representation of tasks, forming clusters that contain embedding vectors of multiple tasks (Figure 2a). Theoretically, this is due to the fact that an accumulative $L^2$ loss of distance between data points is proportional to the dataset variance, which may lead to degenerate distribution such as Bernoulli distribution. This is proven in Appendix B. To build robust and efficient task inference module, we conjecture that it's crucial to ensure *every* task embedding cluster to be separated from each other. We therefore introduce a negative-power variant of contrastive loss as follows:

$$\mathcal{L}_{dml}(\boldsymbol{x_i}, \boldsymbol{x_j}; q) = \mathbf{1}\{y_i = y_j\}||\boldsymbol{q_i} - \boldsymbol{q_j}||^2_2 + \mathbf{1}\{y_i \neq y_j\}\beta \cdot \frac{1}{||\boldsymbol{q_i} - \boldsymbol{q_j}||^n_2 + \epsilon} \quad (13)$$

where $\epsilon > 0$ is a small hyperparameter added to avoid division by zero, the power $n$ can be any non-negative number. Note that when $n = 2$, Eqn 13 takes form analogous to the Cauchy graph embedding introduced by Luo et al. (2011), which was proven to better preserve local topology and similarity relationships compared to Laplacian embeddings. We experimented with 1 (inverse) and 2 (inverse-square) in this paper and compare with the classical $L^1$, $L^2$ metrics in Figure 2 and §5.2.1.

## 5 EXPERIMENTS

In our experiments, we assess the performance of FOCAL by comparing it with several baseline algorithms on meta-RL benchmarks, for which return curves are averaged over 3 random seeds. Specific design choices are examined through 3 ablations and supplementary experiments are provided in Appendix E.

### 5.1 SAMPLE EFFICIENCY AND ASYMPTOTIC PERFORMANCE

We evaluate FOCAL on 6 continuous control meta-environments of robotic locomotion, 4 of which are simulated via the MuJoCo simulator (Todorov et al., 2012), plus variants of a 2D navigation problem called Point-Robot. 4 (Sparse-Point-Robot, Half-Cheetah-Vel, Half-Cheetah-Fwd-Back,

Ant-Fwd-Back) and 2 (Point-Robot-Wind, Walker-2D-Params) environments require adaptation by reward and transition functions respectively.

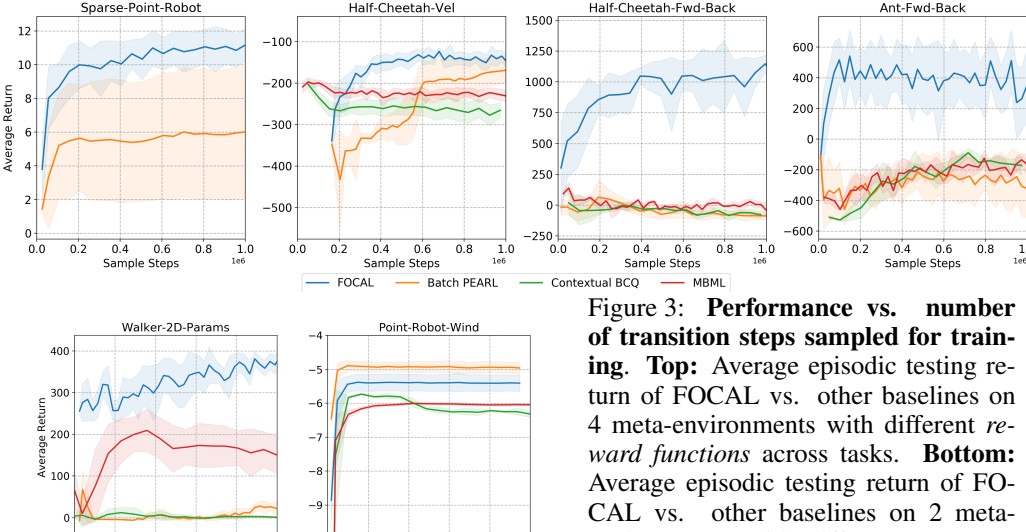

Figure 3: **Performance vs. number of transition steps sampled for training**. **Top:** Average episodic testing return of FOCAL vs. other baselines on 4 meta-environments with different *reward functions* across tasks. **Bottom:** Average episodic testing return of FOCAL vs. other baselines on 2 meta-environments with different *transition dynamics* across tasks.

These meta-RL benchmarks were previously introduced by Finn et al. (2017) and Rakelly et al. (2019), with detailed description in Appendix D. For data generation, we train stochastic SAC models for every single task and roll out policies saved at each checkpoint to collect trajectories. The offline training datasets are selections of the saved trajectories, which facilitates tuning of the performance level and state-action distributions of the datasets for each task.

For OMRL, there are two natural baselines. The first is by naively modifying PEARL to train and test from logged data without exploration, which we term Batch PEARL. The second is Contextual BCQ. It incorporates latent variable $z$ in the state and perform task-augmented variant of offline BCQ algorithm (Fujimoto et al., 2019). Like PEARL, the task inference module is trained using Bellman gradients. Lastly, we include comparison with the MBML algorithm proposed by Li et al. (2019a). Although as discussed earlier, MBML is a model-based, two-stage method as opposed to our model-free and end-to-end approach, we consider it by far the most competitive and related OMRL algorithm to FOCAL, due to the lack of other OMRL methods.

As shown in Figure 3, we observe that FOCAL outperforms other offline meta-RL methods across almost all domains. In Figure 4b, we also compared FOCAL to other algorithm variants including a more competitive variant of Batch PEARL by applying the same behavior regularization. In both trials, FOCAL with our proposed design achieves the best overall sample efficiency and asymptotic performance.

We started experiments with expert-level datasets. However, for some tasks such as Ant and Walker, we observed that a diverse training sets result in a better meta policy (Table 2). We conjecture that mixed datasets, despite sub-optimal actions, provides a broader support for state-action distributions, making it easier for the context encoder to learn the correct correlation between task identity and transition tuples (i.e., transition/reward functions). While using expert trajectories, there might be little overlap between state-action distributions across tasks (Figure 8), which may cause the agent to overfit to spurious correlation. This is the exact problem Li et al. (2019b) aims to address, termed MDP ambiguity. Such overfitting to state-action distributions leads to suboptimal latent representations and poor robustness to distribution shift (Table 6), which can be interpreted as a special form of memorization problem in classical meta-learning (Yin et al., 2019). MDP ambiguity problem is addressed in an extension of FOCAL (Li et al., 2021).

## 5.2 ABLATIONS

Based on our previous analysis, we examine and validate three key design choices of FOCAL by the following ablations. The main results are illustrated in Figure 4 and 5.

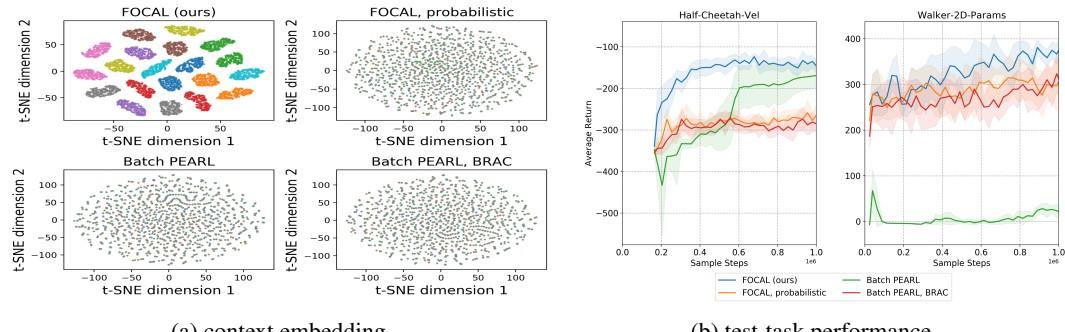

(a) context embedding                    (b) test-task performance

Figure 4: **Comparative study of 4 algorithm variants: FOCAL with deterministic/probabilistic context encoder, Batch PEARL with/without behavior regularization.** (a) t-SNE visualization of the embedding vectors drawn from 20 randomized tasks on Walker-2D-Params. Data points are color-coded according to task identity. (b) Return curves on tasks with different reward functions (Half-Cheetah-Vel) and transition dynamics (Walker-2D-Params).

### 5.2.1 POWER LAW OF DISTANCE METRIC LOSS

To show the effectiveness of our proposed negative-power distance metrics for OMRL problem, we tested context embedding loss with different powers of distance, from $L^{-2}$ to $L^2$. A t-SNE (Van der Maaten & Hinton, 2008) visualization of the high-dimensional embedding space in Figure 2a demonstrates that, distance metric loss with negative powers are more effective in separating embedding vectors of different tasks, whereas positive powers exhibit degenerate behaviors, leading to less robust and effective conditioned policies. By a physical analogy, the inverse-power losses provide "repulsive forces" that drive apart all data points, regardless of the initial distribution. In electromagnetism, consider the latent space as a 3D metal cube and the embedding vectors as positions of "charges" of the same polarity. By Gauss's law, at equilibrium state, all charges are distributed on the surface of the cube with densities positively related to the local curvature of the surface. Indeed, we observe from the "Inverse-square" and "Inverse" trials that almost all vectors are located near the edges of the latent space, with higher concentration around the vertices, which have the highest local curvatures (Figure 7).

To evaluate the effectiveness of different powers of DML loss, we define a metric called *effective separation rate* (ESR) which computes the percentage of embedding vector pairs of different tasks whose distance on latent space $\mathcal{Z}$ is larger than the expectation of randomly distributed vector pairs, i.e., $\sqrt{2l/3}$ on $(-1, 1)^l$. Table 1 demonstrates that DML losses of negative power are more effective in maintaining distance between embeddings of different tasks, while no significant distinction is shown in terms of RMS distance, which is aligned with our insight that RMS or effectively classical $L^2$ objective, can be

Table 1: Embedding Statistics on Half-Cheetah-Vel (latent space dimension $l = 5$).

| Loss | RMS | ESR |
|---|---|---|
| Inverse-square | 1.282 | 0.861 |
| Inverse | 1.217 | 0.840 |
| Linear | 1.385 | 0.819 |
| Square | 1.415 | 0.506 |

optimized by degenerate distributions (Lemma B.1). This is the core challenge addressed by our proposed inverse-power loss.

### 5.2.2 DETERMINISTIC VS. PROBABILISTIC CONTEXT ENCODER

Despite abundance successes of probabilistic/variational inference models in previous work (Kingma & Welling, 2013; Alemi et al., 2016; Rakelly et al., 2019), by comparing FOCAL with deterministic and probabilistic context encoder in Figure 4b, we observe experimentally that the former performs significantly better on tasks differ in either reward or transition dynamics in the fully offline setting. Intuitively, by our design principles, this is due to

1. Offline meta-RL does not require exploration. Also when Assumption 1 is satisfied, there is not need for reasoning about uncertainty during adaption.

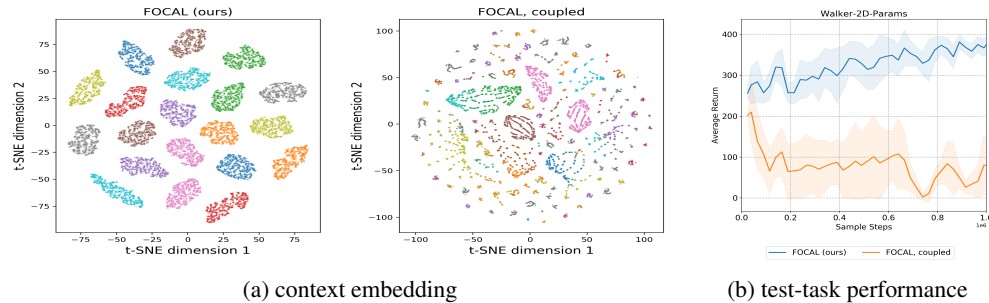

(a) context embedding          (b) test-task performance

Figure 5: **FOCAL vs. FOCAL with coupled gradients.** (a) t-SNE visualization of the embedding vectors drawn from 20 randomized tasks on Walker-2D-Params. Data points are color-coded according to task identity. (b) Return curves on Walker-2D-Params.

    2. The deterministic context encoder in FOCAL is trained with carefully designed metric-based learning objective, detached from the Bellman update, which provides better efficiency and stability for meta-learning.

Moreover, the advantage of our encoder design motivated by Assumption 1 is also reflected in Figure 4a, as our proposed method is the only variant that achieves effective clustering of task embeddings. The connection between context embeddings and RL performance is elaborated in Appendix C.

### 5.2.3 CONTEXT ENCODER TRAINING STRATEGIES

The last design choice of FOCAL is the decoupled training of context encoder and control policy illustrated in Figure 1. To show the necessity of such design, in Figure 4 we compare our proposed FOCAL with a variant by allowing backpropagation of the Bellman gradients to context encoder. Figure 5a shows that our proposed strategy achieves effective clustering of task context and therefore better control policy, whereas training with Bellman gradients cannot. As a consequence, the corresponding performance gap is evident in Figure 5b. We conjecture that on complex tasks where behavior regularization is necessary to ensure convergence, without careful tuning of hyperparameters, the Bellman gradients often dominate over the contribution of the distance metric loss. Eventually, context embedding collapses and fails to learn effective representations.

Additionally however, we observed that some design choices of the behavior regularization, particularly the value penalty and policy regularization in BRAC (Wu et al., 2019) can substantially affect the optimal training strategy. We provide more detailed discussion in Appendix E.2.

## 6 CONCLUSION

In this paper, we propose a novel fully-offline meta-RL algorithm, FOCAL, in pursuit of more practical RL. Our method involves distance metric learning of a deterministic context encoder for efficient task inference, combined with an actor-critic apparatus with behavior regularization to effectively learn from static data. By re-formulating the meta-RL tasks as task-augmented MDPs under the task-transition correspondence assumption, we shed light on the effectiveness of our design choices in both theory and experiments. Our approach achieves superior performance compared to existing OMRL algorithms on a diverse set of continuous control meta-RL domains. Despite the success, the strong assumption we made regarding task inference from transitions can potentially limit FOCAL's robustness to common challenges in meta-RL such as distribution shift, sparse reward and stochastic environments, which opens up avenues for future work of more advanced OMRL algorithms.

## 7 ACKNOWLEDGEMENTS

The authors are grateful to Yao Yao, Zhicheng An and Yuanhao Huang for running part of the baseline experiments. A special thank to Yu Rong and Peilin Zhao for providing insightful comments and being helpful during the working process.

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

# Appendices

## A  PSEUDO-CODE

---

**Algorithm 1:** FOCAL Meta-training

**Given:**

- Pre-collected batch $\mathcal{D}_i = \{(s_{i,j}, a_{i,j}, s'_{i,j}, r_{i,j})\}_{j:1...N}$ of a set of training tasks $\{\mathcal{T}_i\}_{i=1...n}$ drawn from $p(\mathcal{T})$
- Learning rates $\alpha_1, \alpha_2, \alpha_3$

1 Initialize context replay buffer $\mathcal{C}_i$ for each task $\mathcal{T}_i$
2 Initialize inference network $q_\phi(z|c)$, learning policy $\pi_\theta(a|s,z)$ and Q-network $Q_\psi(s,z,a)$ with parameters $\phi$, $\theta$ and $\psi$
3 **while** not done **do**
4   **for** each $\mathcal{T}_i$ **do**
5     **for** t = 0, $T-1$ **do**
6       Sample mini-batches of $B$ transitions $\{(s_{i,t}, a_{i,t}, s'_{i,t}, r_{i,t})\}_{t:1...B} \sim \mathcal{D}_i$ and update $\mathcal{C}_i$
7     **end**
8   **end**
9   Sample mini-batches of $M$ tasks $\sim p(\mathcal{T})$
10   **for** step in training steps **do**
11     **for** each $\mathcal{T}_i$ **do**
12       Sample mini-batches $c_i$ and $b_i \sim \mathcal{C}_i$ for context encoder and policy training
13       **for** each $\mathcal{T}_j$ **do**
14         Sample mini-batches $c_j$ from $\mathcal{C}_j$
15         $\mathcal{L}^{ij}_{dml} = \mathcal{L}_{dml}(c_i, c_j; q)$
16       **end**
17       $\mathcal{L}^i_{actor} = \mathcal{L}_{actor}(b_i, q(c_i))$
18       $\mathcal{L}^i_{critic} = \mathcal{L}_{critic}(b_i, q(c_i))$
19     **end**
20     $\phi \leftarrow \phi - \alpha_1 \nabla_\phi \sum_{ij} \mathcal{L}^{ij}_{dml}$
21     $\theta \leftarrow \theta - \alpha_2 \nabla_\theta \sum_i \mathcal{L}^i_{actor}$
22     $\psi \leftarrow \psi - \alpha_3 \nabla_\psi \sum_i \mathcal{L}^i_{critic}$
23   **end**
24 **end**

---

**Algorithm 2:** FOCAL Meta-testing

**Given:**

- Pre-collected batch $\mathcal{D}_{i'} = \{(s_{i',j'}, a_{i',j'}, s'_{i',j'}, r_{i',j'})\}_{j':1...M}$ of a set of testing tasks $\{\mathcal{T}_{i'}\}_{i'=1...m}$ drawn from $p(\mathcal{T})$

1 Initialize context replay buffer $\mathcal{C}_{i'}$ for each task $\mathcal{T}_i$
2 **for** each $\mathcal{T}_{i'}$ **do**
3   **for** t = 0, $T-1$ **do**
4     Sample mini-batches of $B$ transitions $c_{i'} = \{(s_{i',t}, a_{i',t}, s'_{i',t}, r_{i',t})\}_{t:1...B} \sim \mathcal{D}_{i'}$ and update $\mathcal{C}_{i'}$
5     Compute $z_{i'} = q_\phi(c_{i'})$
6     Roll out policy $\pi_\theta(a|s, z_{i'})$ for evaluation
7   **end**
8 **end**

---

# B    DEFINITIONS AND PROOFS

**Lemma B.1.** *The contrastive loss of a given dataset $\mathcal{X} = \{x_i | i = 1, ..., N\}$ is proportional to the variance of the random variable $X \sim \mathcal{X}$*

*Proof.* Consider the contrastive loss $\sum_{i \neq j} (x_i - x_j)^2$, which consists of $N(N-1)$ pairs of different samples $(x_i, x_j)$ drawn from $\mathcal{X}$. It can be written as

$$\sum_{i \neq j} (x_i - x_j)^2 = 2 \left( (N-1) \sum_i x_i^2 - \sum_{i \neq j} x_i x_j \right) \tag{14}$$

The variance of $X \sim \mathcal{X}$ is expressed as

$$\mathrm{Var}(X) = \overline{(X - \overline{X})^2} \tag{15}$$

$$= \overline{X^2} - (\overline{X})^2 \tag{16}$$

$$= \frac{1}{N} \sum_i x_i^2 - \frac{1}{N^2} (\sum_i x_i)^2 \tag{17}$$

$$= \frac{1}{N^2} \left( (N-1) \sum_i x_i^2 - \sum_{i \neq j} x_i x_j \right) \tag{18}$$

where $\overline{X}$ denotes the expectation of $X$. By substituting Eqn 18 into 14, we have

$$\sum_{i \neq j} (x_i - x_j)^2 = 2N^2 (\mathrm{Var}(X)) \tag{19}$$

$\square$

**Definition B.1** (Task-Augmented MDP). *A task-augmented Markov Decision Process (TA-MDP) can be modeled as $\mathcal{M} = (\mathcal{S}, \mathcal{Z}, \mathcal{A}, P, R, \rho_0, \gamma)$ where*

- *$\mathcal{S}$: state space*
- *$\mathcal{Z}$: contextual latent space*
- *$\mathcal{A}$: action space*
- *$P$: transition function $P(s', z'|s, z, a) = P_z(s'|s, a)$ if there is no intra-task transition*
- *$R$: reward function $R(s, z, a) = R_z(s, a)$*
- *$\rho_0(s, z)$: joint initial state and task distribution*
- *$\gamma \in (0, 1)$: discount factor*

**Definition B.2.** *The Bellman optimality operator $\mathcal{B}_z$ on TA-MDP is defined as*

$$(\mathcal{B}_z \hat{Q})(s, z, a) := R(s, z, a) + \gamma \mathbb{E}_{P(s', z'|s, z, a)} [\max_{a'} \hat{Q}(s', z', a')] \tag{20}$$

**Definition B.3** (Deterministic MDP). *For a deterministic MDP, a transition map $t : \mathcal{S} \times \mathcal{A} \to \mathcal{S}$ exists such that:*

$$P(s'|s, a) = \delta(s' - t(s, a)) \tag{21}$$

where $\delta(x - y)$ is the Dirac delta function that is zero almost everywhere except $x = y$.

## C  IMPORTANCE OF DISTANCE METRIC LEARNING FOR META-RL ON TASK-AUGMENTED MDPs

We provide an informal argument that enforcing distance metric learning (DML) is crucial for meta-RL on task-augmented MDPs (TA-MDPs). Consider a classical *continuous* neural network $N_\theta$ parametrized by $\theta$ with $L \in \mathbb{N}$ layers, $n_l \in \mathbb{N}$ many nodes at the $l$-th hidden layer for $l = 1, ..., L$, input dimension $n_0$, output dimension $n_{L+1}$ and nonlinear continuous activation function $\sigma : \mathbb{R} \to \mathbb{R}$. It can be expressed as

$$N_\theta(\boldsymbol{x}) := A_{L+1} \circ \sigma_L \circ A_L \circ \cdots \circ \sigma_1 \circ A_1(\boldsymbol{x}) \tag{22}$$

where $A_l : \mathbb{R}^{n_{l-1}} \to \mathbb{R}^{n_l}$ is an affine linear map defined by $A_l(\boldsymbol{x}) = \boldsymbol{W}_l x + \boldsymbol{b}_l$ for $n_l \times n_{l-1}$ dimensional weight matrix $\boldsymbol{W}_l$ and $n_l$ dimensional bias vector $\boldsymbol{b}_l$ and $\sigma_l : \mathbb{R}^{n_l} \to \mathbb{R}^{n_l}$ is an element-wise nonlinear continuous activation map defined by $\sigma_l(\boldsymbol{z}) := (\sigma(z_1), ..., \sigma(z_{n_l}))^\mathsf{T}$. Since every affine and activation map is continuous, their composition $N_\theta$ is also continuous, which means by definition of continuity:

$$\forall \epsilon > 0, \quad \exists \eta > 0 \quad \text{s.t.} \tag{23}$$

$$|x_1 - x_2| < \eta \Rightarrow |N_\theta(x_1) - N_\theta(x_2)| < \epsilon \tag{24}$$

where $|\cdot|$ in principle denotes any valid metric defined on Euclidean space $\mathbb{R}^{n_0}$. A classical example is the Euclidean distance.

Now consider $N_\theta$ as the value function on TA-MDP with deterministic embedding, approximated by a neural network parameterized by $\theta$:

$$\hat{Q}_\theta(s, a, z) \approx Q_\theta(s, a, z) = R_z(s, a) + \gamma \mathbb{E}_{s' \sim P_z(s'|s,a)}[V_\theta(s')] \tag{25}$$

The continuity of neural network implies that for a pair of sufficiently close embedding vectors $(z_i, z_j)$, there exists sufficiently small $\eta > 0$ and $\epsilon > 0$ that

$$z_1, z_2 \in \mathcal{Z}, |z_1 - z_2| < \eta \Rightarrow |\hat{Q}_\theta(s, a, z_1) - \hat{Q}_\theta(s, a, z_2)| < \epsilon \tag{26}$$

Eqn 26 implies that for a pair of different tasks $(\mathcal{T}_i, \mathcal{T}_j) \sim p(\mathcal{T})$, if their embedding vectors are sufficiently close in the latent space $\mathcal{Z}$, the mapped values of meta-learned functions approximated by continuous neural networks are sufficiently close too. Since by Eqn 25, due to different transition functions $P_{z_i}(s'|s,a)$, $P_{z_j}(s'|s,a)$ and reward functions $R_{z_i}(s,a)$, $R_{z_j}(s,a)$ of $(\mathcal{T}_i, \mathcal{T}_j)$, the distance between the **true values** of two Q-functions $|Q_\theta(s, a, z_i) - Q_\theta(s, a, z_j)|$ is not guaranteed to be small. This suggests that a meta-RL algorithm with suboptimal representation of context embedding $z = q_\phi(c)$, which fails in maintaining effective distance between two distinct tasks $\mathcal{T}_i, \mathcal{T}_j$, is unlikely to accurately learn the value functions (or any policy-related functions) for both tasks simultaneously. The conclusion can be naturally generalized to the multi-task meta-RL setting.

# D EXPERIMENTAL DETAILS

## D.1 DETAILS OF THE MAIN EXPERIMENTAL RESULT (FIGURE 3 AND 4)

The main experimental result in the paper is the comparative study of performance of FOCAL and three baseline OMRL algorithms: Batch PEARL, Contextual BCQ and MBML, shown in Figure 3. Here in Figure 6 we plot the same data for the full number of steps sampled in our experiments. Some of the baseline experiments only lasted for $10^6$ steps due to limited computational budget, but are sufficient to support the claims made in the main text. We directly adopted the Contextual BCQ and MBML implementation from MBML's official source code[2] and perform the experiments on our own dataset generated by SAC algorithm[3] The DML loss used in experiments in Figure 3 is inverse-squared, which gives the best performance among the four power laws we experimented with in Figure 2.

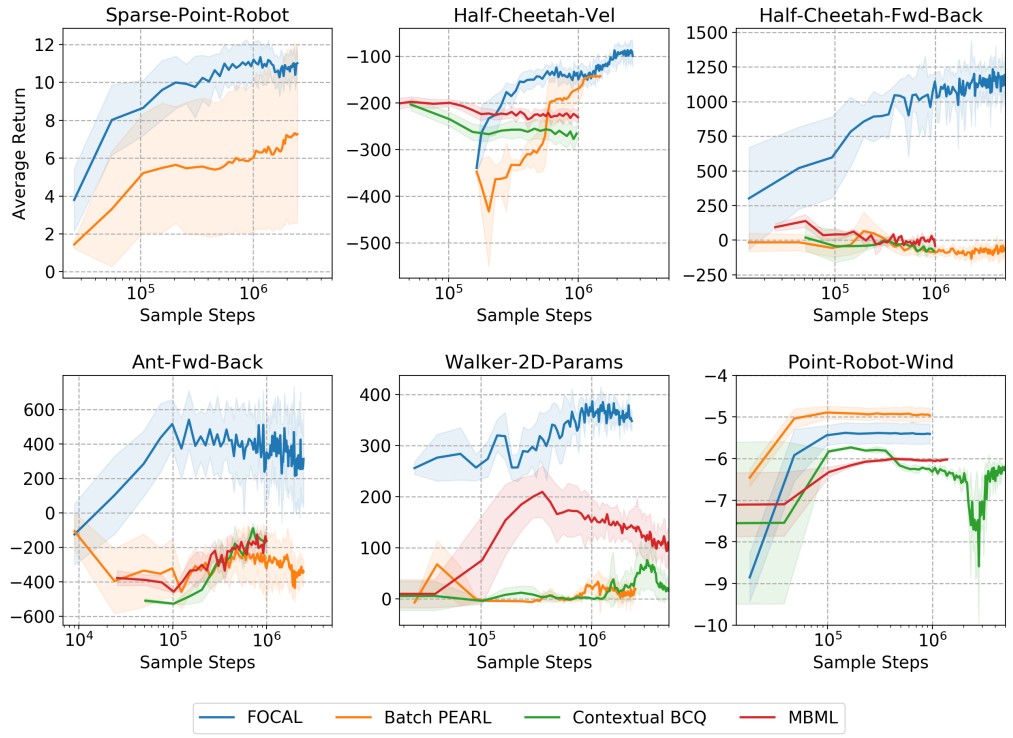

Figure 6: Average episodic testing return of FOCAL vs. other baselines on five meta-environments.

In addition, we provide details on the offline datasets used to produce the result. The performance levels of the training/testing data for the experiments are given in Table 2, which are selected for the best test-time performance over four levels: expert, medium, random, mixed (consist of all logged trajectories of trained SAC models from beginning (random quality) to end (expert quality)). For mixed data, the diversity of samples is optimal but the average performance level is lower than expert. A summary of the fixed datasets used for producing Figure 3 and 6 is given in Table 3.

Lastly, shown in in Figure 7, we also present a faithful 3D projection (not processed by t-SNE) of latent embeddings in Figure 4a. Evidently, our proposed method is the only algorithm which achieves effective clustering of different task embeddings. As validation of our intuition about the analogy between the DML loss and electromagnetism discussed in §5.2.1, the learned embeddings do clus-

---

[2]https://github.com/Ji4chenLi/Multi-Task-Batch-RL

[3]For sparse reward environments like Sparse-Point-Robot, we observed no adapation at test time (return stays zero) for both Contextual BCQ and MBML, which might be due to incorrect implementation or just that both algorithms fail to adapt in sparse reward scenarios. To avoid drawing conlusion too hastily, we chose to not present Contextual BCQ and MBML result for Sparse-Point-Robot at the moment.

Table 2: Quality of data used for best test-time performance. We maintain the same quality of data for training and testing due to algorithm's sensitivity to distribution shift. From our experiments, we observe that for some envs/tasks, datasets with the best performance generate the best testing result, whereas for some envs/tasks, the diversity of data matters the most.

| Meta Env | Training Data | Testing Data |
|---|---|---|
| Sparse-Point-Robot | expert | expert |
| Half-Cheetah-Vel | expert | expert |
| Ant-Fwd-Back | mixed | mixed |
| Half-Cheetah-Fwd-Back | mixed | mixed |
| Walker-2D-Params | mixed | mixed |
| Point-Robot-Wind | expert | expert |

Table 3: Details of the fixed datasets used for producing Figure 3 and 6. The three numbers in the "Checkpoints" column stand for starting epoch: ending epoch: checkpoint spacing.

| Meta Env | # of tasks | Checkpoints | # of trajectories | Trajectory Steps | Obs Dim | Act Dim | Dataset Size |
|---|---|---|---|---|---|---|---|
| Sparse-Point-Robot | 100 | 2200:4800:200 | 51 | 200 | 2 | 2 | 31G |
| Half-Cheetah-Vel | 100 | 50000:950000:50000 | 53 | 1000 | 20 | 6 | 36G |
| Ant-Fwd-Back | 2 | 10000:790000:10000 | 55 | 200 | 27 | 8 | 1.1G |
| Half-Cheetah-Fwd-Back | 2 | 5000:640000:5000 | 6 | 1000 | 20 | 6 | 554M |
| Walker-2D-Params | 50 | 50000:950000:50000 | 53 | 200 | 17 | 6 | 4.6G |
| Point-Robot-Wind | 50 | 2200:11800:200 | 51 | 200 | 2 | 2 | 54G |

ter around the corners and edges of the bounded 3D-projected latent space, which are locations of highest local curvatures.

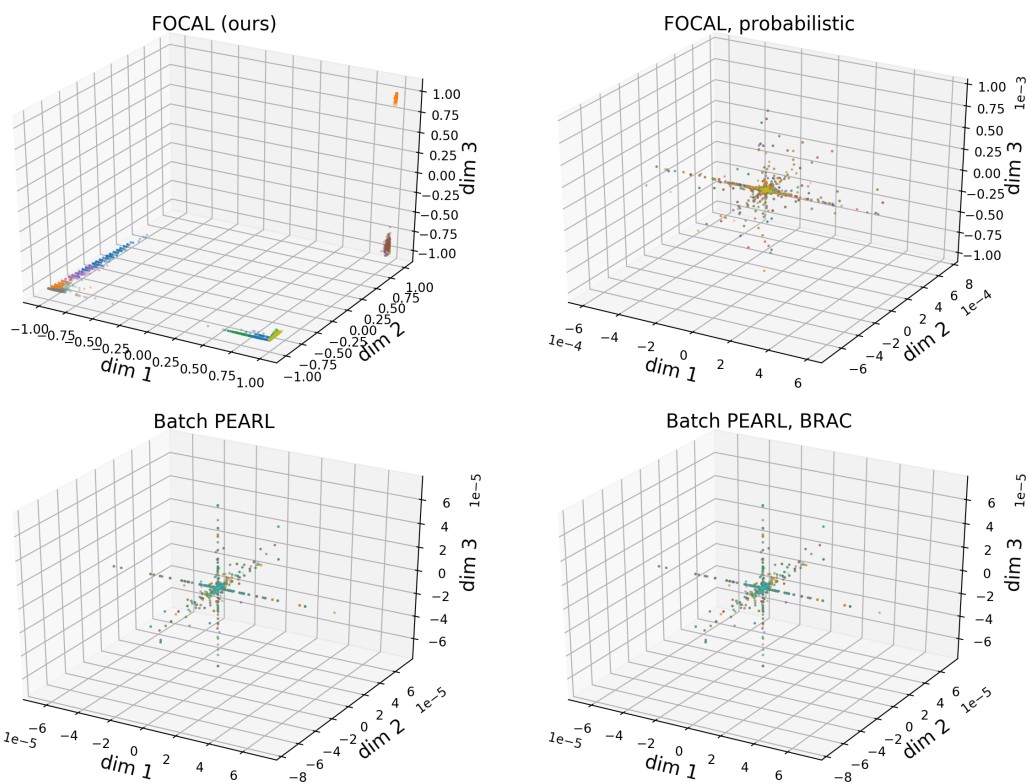

Figure 7: 3D projection of the embedding vectors $\in (-1, 1)^l$ drawn from 20 randomized tasks on Walker-2D-Params. Data points are color-coded according to task identity.

## D.2 Description of the Meta Environments

- **Sparse-Point-Robot**: A 2D navigation problem introduced in PEARL (Rakelly et al., 2019). Starting from the origin, each task is to guide the agent to a specific goal located on the unit circle centered at the origin. Non-sparse reward is defined as the negative distance from the current location to the goal. In sparse-reward scenario, reward is truncated to 0 when the agent is outside a neighborhood of the goal controlled by the goal radius. While inside the neighborhood, agent is rewarded by $1 - $ distance at each step, which is a positive value.

- **Point-Robot-Wind**: A variant of Sparse-Point-Robot. Task **differ only in transition function**. Each task is associated with the same reward but a distinct "wind" sampled uniformly from $[-l, l]^2$. Every time the agent takes a step, it drifts by the wind vector. We use $l = 0.05$ in this paper.

- **Half-Cheetah-Fwd-Back**: Control a Cheetah robot to move forward or backward. **Reward function** is dependent on the walking direction.

- **Half-Cheetah-Vel**: Control a Cheetah robot to achieve a target velocity running forward. **Reward function** is dependent on the target velocity.

- **Ant-Fwd-Back**: Control an Ant robot to move forward or backward. **Reward function** is dependent on the walking direction.

- **Walker-2D-Params**: Agent is initialized with some system dynamics parameters randomized and must move forward, it is a unique environment compared to other MuJoCo environments since tasks differ in transition function. **Transitions function** is dependent on randomized task-specific parameters such as mass, inertia and friction coefficients.

## D.3 Hyperparameter Settings

The details of important hyperparameters used to produce the experimental results in the paper are presented in Table 4 and 5.

Table 4: Hyperparameters used to produce Figure 3. Meta batch size refers to the number of tasks used for computing the DML loss $\mathcal{L}_{dml}^{ij}$ at a time. Larger meta batch size leads to faster convergence but requires greater computing power.

| Hyperparameters | Sparse-Point-Robot | Point-Robot-Wind | Half-Cheetah-Vel | Ant-Fwd-Back | Half-Cheetah-Fwd-Back | Walker-2D-Params |
|---|---|---|---|---|---|---|
| reward scale | 100 | 100 | 5 | 5 | 5 | 5 |
| DML loss weight($\beta$) | 1 | 1 | 10 | 1 | 1 | 10 |
| behavior regularization strength($\alpha$) | 0 | 0 | 50 | 1e6 | 500 | 50 |
| value penalty (in BRAC) | N/A | N/A | False | True | True | False |
| buffer size (per task) | 1e4 | 1e4 | 1e4 | 1e4 | 1e4 | 1e4 |
| batch size | 256 | 256 | 256 | 256 | 256 | 256 |
| meta batch size | 16 | 16 | 16 | 4 | 4 | 16 |
| g_lr(f-divergence discriminator) | 1e-4 | 1e-4 | 1e-4 | 1e-4 | 1e-4 | 1e-4 |
| dml_lr($\alpha_1$) | 1e-3 | 1e-3 | 1e-3 | 1e-3 | 1e-3 | 1e-3 |
| actor_lr($\alpha_2$) | 1e-3 | 1e-3 | 1e-3 | 1e-3 | 1e-3 | 1e-3 |
| critic_lr($\alpha_3$) | 1e-3 | 1e-3 | 1e-3 | 1e-3 | 1e-3 | 1e-3 |
| discount factor | 0.9 | 0.9 | 0.99 | 0.99 | 0.99 | 0.99 |
| # training tasks | 80 | 40 | 80 | 2 | 2 | 20 |
| # testing tasks | 20 | 10 | 20 | 2 | 2 | 5 |
| goal radius | 0.2 | N/A | N/A | N/A | N/A | N/A |
| latent space dimension | 5 | 5 | 20 | 20 | 5 | 20 |
| network width (context encoder) | 200 | 200 | 200 | 200 | 200 | 200 |
| network depth (context encoder) | 3 | 3 | 3 | 3 | 3 | 3 |
| network width (others) | 300 | 300 | 300 | 300 | 300 | 300 |
| network depth (others) | 3 | 3 | 3 | 3 | 3 | 3 |
| maximum episode length | 20 | 20 | 200 | 200 | 200 | 200 |

Table 5: Hyperparameters used to produce Figure 2a

(a) Compared to Half-Cheetah-Vel experiment in Table 4, latent space dimension were reduced to speed up computation. Also the value penalty is used in behavior regularization.

| Hyperparameters | Half-Cheetah-Vel |
|---|---|
| reward scale | 5 |
| behavior regularization strength($\alpha$) | 500 |
| value penalty (in BRAC) | True |
| buffer size (per task) | 1e4 |
| batch size | 256 |
| meta batch size | 16 |
| g_lr(f-divergence discriminator) | 1e-4 |
| dml_lr($\alpha_1$) | 1e-3 |
| actor_lr($\alpha_2$) | 1e-3 |
| critic_lr($\alpha_3$) | 1e-3 |
| discount factor | 0.99 |
| # training tasks | 80 |
| # testing tasks | 20 |
| latent space dimension | 5 |
| network width (context encoder) | 200 |
| network depth (context encoder) | 3 |
| network width (others) | 300 |
| network depth (others) | 3 |
| maximum episode length | 200 |

(b) The DML loss weight $\beta$ and coefficient $\epsilon$ (defined in Eqn 13) used in experiments of Figure 2a to match the scale of objective functions of different power laws. The weights are chosen such that all terms are equal when the average distance of $x_i$ and $x_j$ per dimension is 0.5, a reasonable value given $x \in (-1, 1)^l$.

| Trials | $\beta$ | $\epsilon$ |
|---|---|---|
| Inverse-Square | 1 | 0.1 |
| Inverse | 2 | 0.1 |
| Linear | 8 | 0.1 |
| Square | 16 | 0.1 |

# E    ADDITIONAL EXPERIMENTS

## E.1    SENSITIVITY TO DISTRIBUTION SHIFT

Since in OMRL, all datasets are static and fixed, many challenges from classical supervised learning such as over-fitting exist. By developing FOCAL, we are also interested in its sensitivity to distribution shift for better understanding of OMRL algorithms. Since for each task $\mathcal{T}_i$, our data-generating behavior policies $\beta_i(a|s)$ are trained from random to expert level, we select three performance levels (expert, medium, random) of datasets to study how combinations of training/testing sets with different qualities/distributions affect performance. An illustration of the three quality levels on Sparse-Point-Robot is shown in Fig 8.

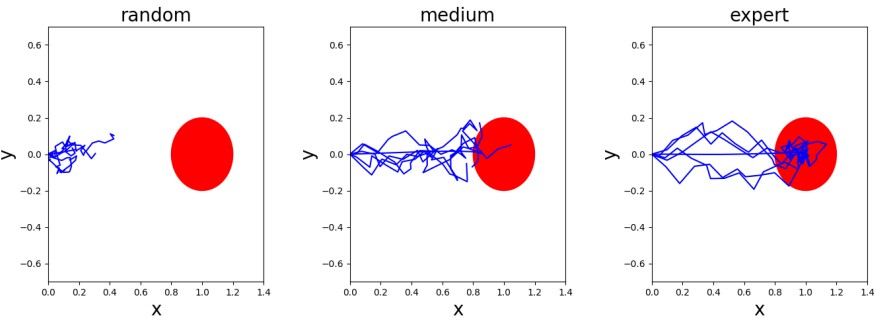

Figure 8: Distribution of rollout trajectories of trained SAC policies of three performance levels: random, medium and expert. Since reward is sparse, only states that lie in the red circle are given non-zero rewards, making meta-learning more challenging and sensitive to data distributions.

Table 6: Average testing return of FOCAL on Sparse-Point-Robot tasks with different qualities/distributions of training/testing sets. The numbers in parenthesis are the performance drop due to distribution shift (compared to the scenario where the testing distribution equals the training distribution).

| Training | Testing | FOCAL$_{(drop)}$ |
|----------|---------|------------------|
| expert   | expert  | $8.16_{(-)}$     |
| medium   | medium  | $8.44_{(-)}$     |
| random   | random  | $2.34_{(-)}$     |
| expert   | medium  | $7.12_{(1.04)}$  |
| expert   | random  | $4.43_{(3.73)}$  |
| medium   | expert  | $8.25_{(0.19)}$  |
| medium   | random  | $6.76_{(1.68)}$  |

Table 6 shows the average return at test-time for various training and testing distributions. Sensitivity to distribution shift is confirmed since training/testing on the similar distribution of data result in relatively higher performance. In particular, this is significant in sparse reward scenario since Assumption 1 is no longer satisfied. With severe over-fitting and the MDP ambiguity problem elaborated in the last paragraph of §5.1, performance of meta-RL policy is inevitably compromised by distribution mismatch between training/testing datasets.

## E.2    VALUE PENALTY AND POLICY REGULARIZATION IN BRAC

Discussed in §3.2, BRAC (Wu et al., 2019) introduces possible regularization in the value/Q-function (Eqn 6/7) and therefore the critic loss (Eqn 8), as well as in the actor loss (Eqn 9). If regularization is applied on both or only on the policy, it is referred to as value penalty and policy regularization respectively. In the BRAC paper, Wu et al. (2019) performed extensive tests and concluded that the two designs yield similar performance, with value penalty being slightly better overall. Since BRAC is designed for single-task offline RL, we again tested both on our OMRL setting. In general, we

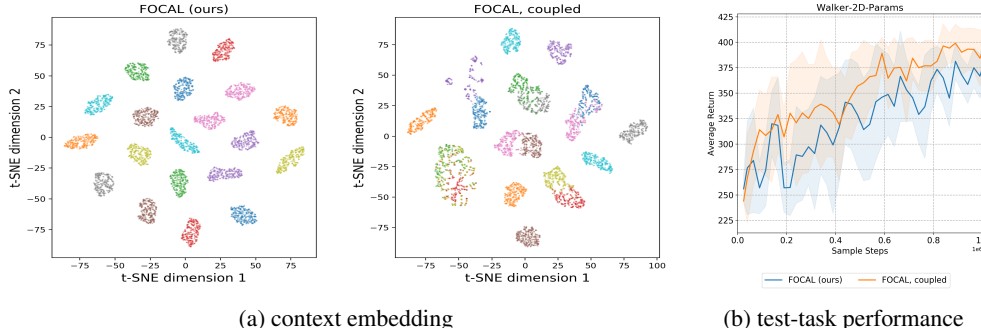

(a) context embedding

(b) test-task performance

Figure 9: **FOCAL vs. FOCAL with coupled gradients and policy regularization. The task representation alone of the coupled training scheme might not be superior, but the policy performance can be improved due to end-to-end optimization.** (a) t-SNE visualization of the embedding vectors drawn from 20 randomized tasks on Walker-2D-Params. Data points are color-coded according to task identity. (b) Return curves on Walker-2D-Params.

found that on complex tasks such as Ant, value penalty usually requires extremely large regularization strength (Table 4) to converge. Since the regularization is added to the value/Q-function, this results in very large nagative Q value (Figure 10) and exploding Bellman gradients. In this scenario, training the context embedding with backpropogated Bellman gradients often yields sub-optimal latent representation and policy performance (Fig 5), which leads to our design of decoupled training strategy discussed in §5.2.3.

For policy regularization however, the learned value/Q-function approximates the real value (Figure 11a), leading to comparable order of magnitude for the three losses $\mathcal{L}_{dml}$, $\mathcal{L}_{actor}$ and $\mathcal{L}_{critic}$. In this case, the decoupled training of context encoder, actor and critic, may give competitive or even better performance due to end-to-end optimization, shown in Figure 9.

### E.3 DIVERGENCE OF Q-FUNCTIONS IN OFFLINE SETTING

The necessity of applying behavior regularization on environment like Ant-Fwd-Back and Walker-2D-Params to prevent divergence of value functions is demonstrated in Figure 10 and 11.

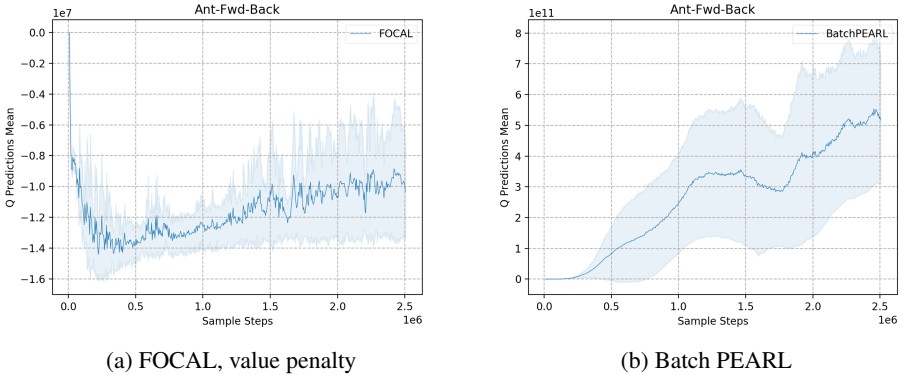

(a) FOCAL, value penalty

(b) Batch PEARL

Figure 10: **FOCAL with value penalty vs. Batch PEARL on Ant-Fwd-Back.** The Q-function learned by Batch PEARL diverges ($> 10^{11}$) whereas the Q-function of FOCAL, despite its large order of magnitude due to **value penalty**, converges eventually given proper regularization ($\alpha = 10^6$)

.

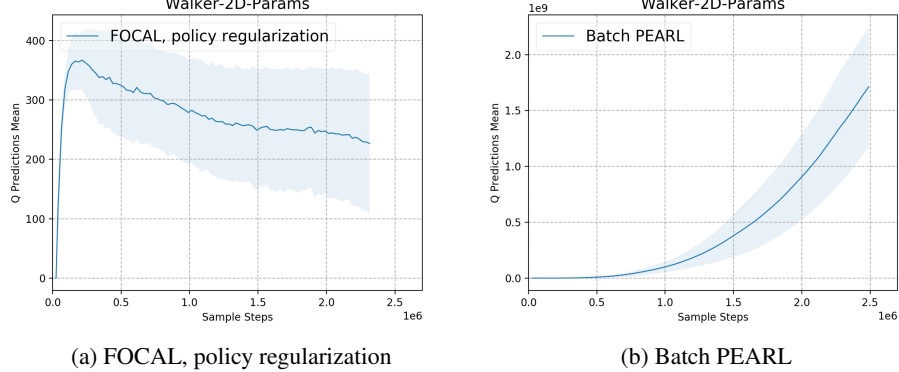

(a) FOCAL, policy regularization

(b) Batch PEARL

Figure 11: **FOCAL with policy regularization vs. Batch PEARL on Walker-2D-Params.** The Q-function learned by Batch PEARL diverges ($> 10^8$) whereas the Q-function of FOCAL, converges to the true discounted cumulative return ($\approx 200$ for $\gamma = 0.99$).

## F  IMPLEMENTATION

We build our algorithm on top of PEARL and BRAC, both are derivatives of the SAC algorithm. SAC is an off-policy actor-critic method with a maximum entropy RL objective which encourages exploration and learning a stochastic policy. Although exploration is not needed in fully-offline scenarios, we found empirically that a maximum entropy augmentation is still beneficial for OMRL, which is likely due to the fact that in environments such as Ant, different actions result in same next state and reward, which encourages stochastic policy.

All function approximators in FOCAL are implemented as neural networks with MLP structures. For normalization, the last activation layer of context encoder and policy networks are invertible squashing operators (tanh), making $\mathcal{Z}$ a bounded Euclidean space $(-1, 1)^l$, which is reflected in Figure 7.

As in Figure 1, the whole FOCAL pipeline involves three main objectives. The DML loss for training the inference network $q_\phi(z|c)$ is given by Eqn 13, for mini-batches of transitions drawn from training datasets: $\boldsymbol{x_i} \sim \mathcal{D}_i$, $\boldsymbol{x_j} \sim \mathcal{D}_j$. The embedding vector $\boldsymbol{q_i}$, $\boldsymbol{q_j}$ are computed as the average embedding over $\boldsymbol{x_i}$ and $\boldsymbol{x_j}$. The actor and critic losses are the task-augmented version of Eqn 8 and 9:

$$\mathcal{L}_{\text{critic}} = \mathbb{E}_{\substack{(s,a,r,s')\sim\mathcal{D} \\ a'\sim\pi_\theta(\cdot|s')}} \left[ \left( r + \gamma \bar{Q}_\psi^D(s', \bar{z}, a') - Q_\psi(s, \bar{z}, a) \right)^2 \right] \tag{27}$$

$$\mathcal{L}_{\text{actor}} = -\mathbb{E}_{(s,a,r,s')\sim\mathcal{D}} \left[ \mathbb{E}_{a''\sim\pi_\theta(\cdot|s)}[Q_\psi(s, \bar{z}, a'')] - \alpha\hat{D} \right] \tag{28}$$

where $\bar{Q}$ is a target network and $\bar{z}$ indicates that gradients are not being computed through it. As discussed in (Kumar et al., 2019; Wu, Tucker, and Nachum, 2019), the divergence function $\hat{D}$ can take form of Kernel MMD (Gretton et al., 2012), Wasserstein Divergence (Arjovsky, Chintala, and Bottou, 2017) or f-divergences (Nowozin et al., 2016) such as KL divergence. In this paper, we use the dual form (Nowozin, Cseke, and Tomioka, 2016) of KL divergence, which learns a discriminator $g$ with minimax optimization to circumvent the need of a cloned policy for density estimation.

In principle, as a core design choice of PEARL, the context used to infer $q_\phi(z|c)$ can be sampled with a different strategy than the data used to compute the actor-critic losses. In OMRL however, we found this treatment unnecessary since there is no exploration. Therefore training of DML and actor-critic objectives are randomly sampled from the same dataset, which form an end-to-end algorithm described in Algorithm 1 and 2.

