# OpenReview forum: "FOCAL: Efficient Fully-Offline Meta-Reinforcement Learning via Distance Metric Learning and Behavior Regularization"
_ICLR.cc/2021/Conference — ICLR 2021 Poster_

### Official Review · AnonReviewer2 · 2020-10-28

**Rating:** 7
**Confidence:** 3

**Review:**

This paper tackles the problem of offline meta reinforcement learning, where an agent aims to learn a policy which can adapt to an unseen task (dynamics/reward), but from entirely offline data. As a result of being fully offline, the agent can no longer explore in the new task at test time, but instead receives randomly sampled transitions from the new task, from which it must infer the task. They then propose a method for learning task inference from fully offline data, as well as a policy conditioned on this task encoding from offline data built off behavior regularized actor critic (BRAC). Results indicate that in this outperforms PEARL as well as multi-task offline RL with BCQ.

Pros:
The problem this paper tackles is important, and is formulated in a practical way. While in general meta-RL is often closely tied to online exploration to infer what the task is, in practice it seems like having an agent be able to adapt to a new task given just a few transitions from a potentially different policy would be a valuable instantiation of meta-RL. It also integrates nicely with progress in batch RL.

Second, the idea of decoupling the task inference objective using the metric loss does nicely integrate with the setting of using an offline dataset, and seems to work well as shown by the qualitative examples. While it does depend heavily on having access to the task labels during training, this assumption is used in a lot of meta-RL work. Additionally, the proposed negative power variant of the metric loss does seem to better encourage separation of tasks with different task labels, and seems like a good tool for metric learning in general.

Lastly, the experiments do compare against the relevant baselines to the best of my knowledge, and on tasks similar to those used in prior works. The ablations are also interesting, and make a compelling case for why in the offline case deterministic context encoders may be more fitting (when the tasks satisfy Assumption 1).

Cons:
In general I think the key assumption in this work, that tasks can be inferred from an unordered set of transitions is limiting, especially for tasks with sparse reward, or which involve adapting to changing dynamics. However, this does seem like it is the best that one can do while being totally offline, any more complex tasks would require some form of online exploration.

I'm not convinced of the argument in Appendix C, that the reason why using the control policy to learn the context encoder works poorly is due to neural networks inability to tell apart small differences in the embedding vectors. Rather I think the decoupled training allows for an easier optimization of the task embedding, and then the policy can be learned using good (and stationary) task embeddings throughout all training.

---

> ### Author Response · Authors · 2020-11-23
> **Response to Reviewer 2**
>
> We thank the reviewer for recognition of several key elements of our paper:
>   1. Well-motivated contribution in addressing the important and practical problem of offline meta-RL
>   2. Insight of several key design choices customized for offline meta-RL setting (elaborated in the three ablations):
>
>     i. Deterministic context encoder instead of a probabilistic variant adopted in SOTA baselines such as PEARL
>
>     ii. Novel Negative power variant of the metric loss instead of common L1, L2 objective
>
>     iii. Decoupling the task inference training from control policy using the metric loss
>
>   3. Comprehensive experimental evidence for design choices listed above (section 5.2)
>
> To address your questions:
>
> - "key assumptions is limiting":
>
> First of all, we fully concur that this is a very good point. We realize that in real-world envs, Assumption 1 can almost never be strictly satisfied. **In response, we have added in-depth discussion in 5.1 and 5.2.1, to show that although Assumption seems like a severe constrant to the scope of FOCAL, the experimental evidence demonstrated that the algorithm is robust enough to achieve good performance without strictly satisfying Assumption 1, like on Sparse-Point-Robot with sparse reward for example.**
>
> Formally, we would like to consider Assumption 1 a sufficient condition for the feasibility of our permutation-invariant and deterministic context encoder design. It provides intuitive theoretical insight but is too strong (not necessary) for the algorithm to work well. On the other hand, Assumption is still reasonable for tasks associated with unique reward/transition functions, which covers 4 out of 5 of the meta environments we tested: Half-Cheetah-Vel, Ant-Fwd-Back, Half-Cheetah-Fwd-Back and Walker-2D-Params. Therefore our analyses based on such assumption still has value in its own right.
>
> If we think FOCAL as one step closer to general/robust model-free OMRL algorithm, then this "limiting factor" should be looked more from the bright side. As discussed in the conclusion section, we totally concur that this is exactly the point (or at least one of the most important) where FOCAL can be improved upon in future studies. We consider techniques for more robust task inference, potentially including but not restricted to such as active learning of task labels, attention mechanism of transition sequence or certain stochasticity for quantifying the uncertainty introduced by the confounding samples as possible future directions to improve FOCAL.
>
> - "Not convinced by Appendix C":
>
> We totally agree that the decoupled training, as one of the key design choices of FOCAL (5.2.3, Appendix E.2), makes it easier to learn effective task representation on embedding space. But we think it is compatible with our argument in Appendix C. The point of Appendix C is to show that without effective separation of different tasks realized by technique such as DML, it's difficult or even impossible for neural networks to accurately approximate z-conditioned value functions across multiple tasks simultaneously. The DML and decoupled training are synergetic by design.
>
> Overall we are really grateful for you appreciation of our work and recognizing its key contributions. We hope this response addresses your concerns. Please let us know if you have any further comments or concerns or ways in which we can further improve our paper.

---

### Official Review · AnonReviewer4 · 2020-10-29

**Rating:** 5
**Confidence:** 3

**Review:**

Thanks for the feedback. However, I think most of the performance gain came from behavior regularization, which is an already existing technique, rather than the proposed metric learning method (Figure3 and Figure 5). I’ll keep my ratings unchanged.

============

Summary:
The paper presents a mode-free, end-to-end offline meta-reinforcement learning algorithm.
The authors proposed the context encoder to encode the task from the history of transitions (called the “context”) using metric-based approaches. Specifically, they embedded the context to a latent vector z by clustering similar data points (context) taken from the same task while pushing away dissimilar data points taken from different tasks. The latent context variable z is used to condition the actor and the critic.
Experiments are performed on the MuJoCo simulator and the 2D navigation problem with a sparse reward called Sparse-Point-Robot, using different reward functions depending on the task. The method proposed, FOCAL (Fully-Offline Context-based Actor-critic meta-rL algorithm), outperforms other offline meta-RL algorithms.
The main contribution of the paper comes from combining the context-based approach with metric learning.

Pros:
- Overall, the paper is well written. The method of using metric learning has been well-motivated by toy experiments on the context embedding.
- The paper tackles offline reinforcement learning and meta-learning, which are emerging topics in RL/ML.

Cons:
- Format: The paper violated the paper length limit of 8 pages for the main context.
- Missing experiments: The paper directly followed the algorithm and the experiment setting of “Efficient Off-Policy Meta-RL” (Rakelly et al., 2019) while omitting experiments on the MuJoCo walker and humanoid environments. Also, for the walker-2d experiment, only the toy experiment on context embedding is available with no return curve.
- Ablations: The authors compare their method (FOCAL) to Batch PEARL while describing the Batch PEARL as “a vanilla version of FOCAL without behavior regularization or distance metric learning.” It would be interesting to compare FOCAL and Batch PEARL both with the behavior regularization to observe the effect of metric learning in isolation.
- Novelty: The algorithm in Appendix A seems incremental to the algorithm in “Efficient Off-Policy Meta-RL” (Rakelly et al., 2019). The only difference is that the setting is offline and using metric learning to encode the context instead of KL-divergence. Although the combination of context encoding and metric learning is interesting, the proposed is method is still incremental.


Minor comments:
- There was no source code when I followed the Github link provided in the paper.
- In the return curves on Half-Cheetah-Vel and Ant-Fwd-Back in Figure 3, there is no value near 1e6 sample steps, where the deterioration of performance begins.
- I understood that all tasks share the same state and action space, with different transition and reward functions. But all the experiments in the paper use the same transition. I wonder if there is an experiment with a different transition depending on the task.

---

> ### Author Response · Authors · 2020-11-23
> **Response to Reviewer 4**
>
> We appreciate that the reviewer recoginizes our well-motivated contribution in addressing offline meta-RL problem, as well as clarify in writing.
>
> To address your questions:
>
> - "Format: violation of 8-page limit":
>
> Thank you for bringing this up. We indeed stuggled to fit in all content within the 8-page limit for the first draft and had to put quite a lot in the Appendix. We have strictly sticked to the 9-page rule for the rebuttal version this time.
>
> - "Missing experiments":
>
> This is a good point. **We have now included the experiments on walker in 5.2.3 of the rebuttal version.** The reason we didn't do walker and humanoid (two most complex meta-envs tested by PEARL (Rakelly et al., 2019)) was due to constrained computation budget, and unfortunately within the rebuttal period, we are still short on resources to finish humanoid experiment.  For humanoid, we have already collected over 1TB (100 tasks, over 20 checkpoints, 50 sampled trajectories for each checkpoint) data, unfortunately the complete training/testing procedure is still ongoing. This is something surely we can carefully prepare and add to the camera-ready version should this paper be accepted.
>
> Most importantly, we prioritize walker mainly because it's a unique environment where tasks differ in transition function, whereas all other envs differ in rewards. **With walker included, we think we have now covered all representative cases (sparse reward, different reward/transition functions, binary/near continuous goals) for testing the performance/robustness of FOCAL. So we think the experiments presented in the latest version should be sufficient enough for validation of our algorithm.** Please let us know if you think differently or have any further questions on this. We can experiment with more complex environments for the camera-ready version if the paper is accepted.
>
> - "Ablations":
>
> In fact, we included the said ablation in Appendix E.2 and figure 6 of the first draft. On sparse-point-robot in
> particular, the behavior regularization has little effect and FOCAL still outperforms BatchPEARL by a large margin. It is our fault that this is not in the main text since we really struggled to make space for everything within the page constraint. We now have added such ablation on walker env in 5.2.3, our method FOCAL still clearly outperforms the baseline.
>
> - "Novelty":
>
> We agree that our main framework builds on PEARL by using DML and BRAC, but it is by no means a naive combination of the existing methods. As we stressed in the main text (like last paragraph of section 1, first paragraph of section 4), we consider the novelty/contribution of our method as twofold:
>   1. Proposes the first model-free and end-to-end algorithm that successfully address the important and practical (reviewer quote: "well-motivated") offline meta-RL problem.
>   2. Incorporate several key design choices customized for offline meta-RL setting, all well-motivated and different from SOTA baselines such as PEARL:
>
>     i. Deterministic context encoder instead of a probabilistic variant adopted in SOTA baselines such as PEARL
>
>     ii. Novel Negative power variant of the metric loss instead of common L1, L2 objective
>
>     iii. Decoupling the task inference training from control policy using the metric loss
>
> - minor comments
>
> Also, to address your concerns in the minor comments, we finally obtained permission to publish the code and now it's in the supplementary material. We also recovered some missing data points in Fig 3 from log, and indeed there are moderate performance deterioration on environments like sparse-point-robot and half-cheetah-vel. Accordingly, we have included our insight and discussion regarding this observation in 5.2.1, hope this helps.
>
> Overall, we are really grateful for your comments and advice, which are all valuable in helping us improve the work. Hope our explanation and updated material address your concerns. Please let us know if you have any further comments or concerns or ways in which we can further improve our paper.

---

### Official Review · AnonReviewer3 · 2020-10-30
**Proposes novel algorithm for offline meta-reinforcement learning**

**Rating:** 5
**Confidence:** 4

**Review:**

Summary :
The paper studies meta-reinforcement learning in the fully offline setting, and proposes a novel algorithm 'FOCAL'. Given offline datasets for tasks sampled from some prior, the algorithm learns a context encoder using distance-based metrics. The encoder is used for inferring the task-latent $z$, which is used to condition the policy rollout $\pi(a | s, z)$. They demonstrate experiments where FOCAL outperforms baselines like PEARL.

Reasons for score :
While I concur with the motivations of the paper, I vote for rejecting the paper (marginally below acceptance threshold). My main concern is the limited novelty of the proposed solution and some missing ablations (c.f weaknesses). I encourage the authors to incorporate feedback for the reviewers and work towards a stronger submission.

Strengths:
+ The paper is well written and quite easy to follow. The problem is sufficiently well motivated, and the algorithm builds on previous approaches, particularly PEARL/BRAC.
+ The proposed algorithm seems to outperform reasonable baselines on few navigation/Mujoco tasks in the offline setting.

Weaknesses:
- As the authors identify, the assumption of task-identifiability from any transition (s, a) seems infeasible in a practical setting. This significantly reduces the scope of the algorithm. The novelty of the proposed algorithm is also quite limited it builds on the framework of PEARL by using (i) distance-based metrics for task inference, which have been studied in the multi-task literature (ii) offline policy learning, where it uses off-the-shelf baselines from the offlineRL literature.
- The authors include few ablations, I believe adding one/more of the following would improve the quality of discussion:
       (i) Evaluating how well the context-encoder performs in inferring the meta-test tasks?
       (ii) Ablating the choice of encoder : Comparing the performance of BatchPEARL, where the encoder is still learned with Bellman gradients, while the policy is behaviorally regularized. I consider this to be a stronger baseline than the existing version of BatchPEARL.
- Experiments pertaining to 5.2.2 in the ablations are not well-motivated. Given the contrastive loss definition in equation(13), it follows that the “inverse-power” losses learn reasonable task embeddings. In my understanding, n>0 here invalidates the choice of “linear”/”square” objectives. I’d be happy to be corrected if I misunderstood the setup.
- The authors are encouraged to add details sufficient to reproduce the results. In particular, the setup of the baseline “ContextualBCQ” is scant on details.

Miscellaneous:
- The paper has several minor grammatical errors which could be corrected in the final draft.
- I would encourage the authors to add more discussion to the experiments section. The current version reflects the empirical results, but provides little insight into them. Last half of section 5.1 has some mention on the importance of choice of datasets, but doesn’t provide details (until in the Appendix).

---

> ### Author Response · Authors · 2020-11-23
> **Response to Reviewer 3 - Part 1**
>
> We appreciate that the reviewer recoginizes our well-motivated contribution in addressing offline meta-RL problem, as well as clarify in writing.
>
> To address your questions:
>
> - "key assumption is infeasible in a practical setting":
>
> First of all, we fully concur that this is a very good point. We realize that in real-world envs, Assumption 1 can almost never be strictly satisfied. **In response, we have added in-depth discussion in 5.1 and 5.2.1, to show that although Assumption seems like a severe constrant to the scope of FOCAL, the experimental evidence demonstrated that the algorithm is robust enough to achieve good performance without strictly satisfying Assumption 1, like on Sparse-Point-Robot for example.**
>
> Formally, we would like to consider Assumption 1 a sufficient condition for the feasibility of our permutation-invariant and deterministic context encoder design. It provides intuitive theoretical insight but is too strong (not necessary) for the algorithm to work well. On the other hand, Assumption is still reasonable for tasks associated with unique reward/transition functions, which covers 4 out of 5 of the meta environments we tested: Half-Cheetah-Vel, Ant-Fwd-Back, Half-Cheetah-Fwd-Back and Walker-2D-Params. Therefore our analyses based on such assumption still has value in its own right.
>
> If we think FOCAL as one step closer to general/robust model-free OMRL algorithm, then this "limiting factor" should be looked more from the bright side. As discussed in the conclusion section, we totally concur that this is exactly the point (or at least one of the most important) where FOCAL can be improved upon in future studies. We consider techniques for more robust task inference, potentially including but not restricted to such as active learning of task labels, attention mechanism of transition sequence or certain stochasticity for quantifying the uncertainty introduced by the confounding samples as possible future directions to improve FOCAL.
>
> - "Novelty":
>
> We agree that our main framework builds on PEARL by using DML and BRAC, but it is by no means a naive combination of the existing methods. As we stressed in the main text (like last paragraph of section 1, first paragraph of section 4), we consider the novelty/contribution of our method as twofold:
>
>   1. Proposes the first model-free and end-to-end algorithm that successfully address the important and practical (reviewer quote: "well-motivated") offline meta-RL problem.
>
>   2. Incorporate several key design choices customized for offline meta-RL setting, all well-motivated and different from SOTA baselines such as PEARL:
>
>     i. Deterministic context encoder instead of a probabilistic variant adopted in SOTA baselines such as PEARL
>
>     ii. Novel Negative power variant of the metric loss instead of common L1, L2 objective
>
>     iii. Decoupling the task inference training from control policy using the metric loss
>
> - "More Ablations":
>
>     1. "Evaluating how well the context-encoder performs in inferring the meta-test tasks? "
>
>     We have provided evaluation of context-encoder for task inference with both visualization and statistics, not specifically but along with ablations. Figure 4(a), 5(a) illustrate the test-performance of clustering of embedding vectors generated by the proposed encoder compared to other design choices. Table 1 shows the test-performance effective separation rate (ESR) across different distance metrics, among which our proposed encoder design performs the best. In our opinion, these are direct evidence of "how well the context-encoder performs in inferring the meta-test tasks". We also provided theoretical insight on how effective clustering of embedding vectors enables better approximation of values functions by continuous neural networks in Appendix C. We woud be glad to provide further clarification or evidence if there is any misunderstanding.
>
>     2. "Ablating the choice of encoder"
>
>     This is a good point. In fact, we already included the said ablation in Appendix E.2 and figure 6 of the first draft. On sparse-point-robot in particular, the behavior regularization has little effect and FOCAL still outperforms BatchPEARL by a large margin. We are sorry that these were not in the main text since we really struggled to make space for everything within the page constraint. **For rebuttal revision, we have now added such ablation on walker env in 5.2.3, FOCAL still clearly outperforms the baseline.**
>
> - "Ablations in 5.2.2 (power law of distance metric loss) not well-motivated":
>
> The point of 5.2.2 is to demonstrate one of our key design choices or novelty points, that using carefully designed metrics such as "negative-power" losses are superior than naive L1, L2 losses (contrastive loss in (12)) for task inference in meta-RL, which are positive powers. As a result of this study, we chose negative-power form in (13) over (12). We don't see why this is not well-motivated but are open to further discussion.

---

> > ### Author Response · Authors · 2020-11-23
> > **Response to Reviewer 3 - Part 2**
> >
> > - "Add more details on reproducing the results, such as the setup of 'ContextualBCQ':
> >
> > In the rebuttal version, we describe "ContextualBCQ" as "By incorporating a latent space z in the state space for task inference, it can be interpreted as the task-augmented variant of offline BCQ algorithm (Fujimoto et al., 2019). Like PEARL, the
> > task inference module is trained using Bellman gradients." We directly adopted ContextualBCQ and DistilledBCQ from the cited paper "Multi-task Batch Reinforcement Learning withMetric Learning" (Li et al., 2019b) and implemented the algorithms on our own datasets. Since we directly used the linked source code for reproducing the result in Fig 3, for further information, we would encourage the reviewers to refer to
> > https://papers.nips.cc/paper/2020/file/4496bf24afe7fab6f046bf4923da8de6-Paper.pdf for experimental details. Please let us know if you have further questions on this issue.
> >
> > - miscellaneous (add more discussion in experiments section):
> >
> > This is a good point, we didn't do this in the first draft because we really struggled to make space for everything within the 8-page limit. Now with 1 more page, please check section 5 of the latest version for more discussion (**also our latest updates for all readers**), as well as one more ablation study of FOCAL vs. Batch PEARL + BRAC at the request of reviewr 3 & 4. Also now we have attached the appendix as part of the main pdf, it should be easier for you to link the related content. Please check Appendix D for the experimental details. We found it's the most appropriate place to present them. If you still think anything should be added to the main tex or the appendix, please feel free to  let us know.
> >
> > Overall, we are really grateful for your comments and advice, which are all valuable in helping us improve the work. Hope our explanation and updated material address your concerns. Please let us know if you have any further comments or concerns or ways in which we can further improve our paper.

---

### Author Response · Authors · 2020-11-23
**Annoucement To All Reviewers and AC/PCs**

We really appreciate your efforts in helping us reflect and improve the paper. In light of the reviewers' comments, we made the following major revision based on the previous manuscript (now submitted as the rebuttal version):
1. Added ablation of training context encoder via DML vs. Bellman gradients (equivalent to FOCAL vs. Batch PEARL with BRAC) in 5.2.3, as a stronger baseline compared to Batch PEARL without BRAC
2. Added return curve on walker env also in 5.2.3, where tasks have different transition functions rather than reward functions (as in other envs we experimented with)
2. Added more extensive discussion on the experimental section (section 5), including our insight on some of the failure cases (overfitting , performance deterioration) of FOCAL
4. Updated return curve on sparse-point-robot in Fig 3 with our latest data obtained this weekend, no significant deteriotation this time. We also connected the missing dots on all return curves in Fig 3, from the data we already had before. Especially for FOCAL and Batch PEARL, now the data points should be complete.
5. Uploaded source code and part of the data required for reproducing the key results in main text
6. Attached the appendix to the main text,  should make things easier for readers to read and link relevant content

Thank you all again for your time and valuable advice! We will adresss each reviewer's individual question momentarilly. Feel free to let us know if you have any comments/questions.

---

### Author Response · Authors · 2020-11-25
**Updates**

Dear Readers,

We have updated our final rebuttal version as well as the source code as supplementary material, including the following major revisions:

1. Added full-range return curves for all five meta-envs in Fig 6 (Appendix D.1 ). **On all envs, FOCAL outperforms baselines by a large margin with no significant performance deterioration.**

2. Updated Half-Cheetah-Vel return curve with smaller meta batch size (80 -> 16) in Fig 3, to be consistent with our other experiments. The relevant hyperparameters for reproducing the result is also updated in Table 3. **Like our previous updates on Sparse-Point-Robot, no significant performance deterioration is observed this time**.

3. At request of the reviewers, we further strengthen our discussion in the experiments section. Particullay in 5.1 and 5.2.1, we provide in-depth analyses on characteristics of our offline meta-RL algorithm. Since reviewers are concerned with limitation of our Assumption 1, which we totally concur and provide argument on why this is important. **However, we also stress based on the experimental evidence that overall, FOCAL is able to achieve excellent performance and robustness even when Assumption 1 is not satisfied. We would like to think Assumption 1 more as a sufficient codition and strong abstration which gives intuitive theoretical insight on why FOCAL works well in OMRL setting. But the message we got from the experimental results is that it's not necessary for FOCAL to work well on many (real-world) tasks where Assumption 1 is never strictly satisfied.**

4. Updated description of meta-envs in Appendix D.2 to explain how tasks differ, particularlly in reward or transition functions.

Again thank you all for your valuable input. Since we have not received any feedback regarding our rebuttal so far, we have done our best based on the previous reviews to work towards a stronger submission. Hope the rebuttal version can help make things more clear. We look forward to your comments and final decision.

---

### Decision · Program_Chairs · 2021-01-07
**Final Decision**

**Decision:**

Accept (Poster)

**Comment:**

Meta-learning for offline RL is an understudied topic with lots of potential impact in the research community. This paper takes the first stab against that challenging problem by proposing a solution similar based on PEARL and distance metric learning. The results look good and it seems like the authors have addressed some of the concerns raised by the reviewers. As a result, I suggest to accept this paper.

However, this paper still has some shortcomings as reviewers suggested more baselines with more experiments on standardized benchmarks such as D4RL or RL Unplugged could make the paper stronger.